# DYNAMIC HUMAN AI COLLABORATION

**Parth Pahwa**
Decision Science
JPMorgan Chase

**Kabir Thakur**
School of Information Studies
Syracuse University

**Francois Buet-Golfouse**
Decision Science, JPMorgan Chase
University College London

## ABSTRACT

Domain experts possess valuable knowledge and insights that can help improve the accuracy and relevance of the machine learning (ML) models. By incorporating expert opinions, the models can capture important nuances and factors that may not be captured by data-driven methods alone. The integration of machine learning models with human experts has become increasingly common in real-world applications. In this paper, we propose a Bayesian framework for human-in-the-loop pipelines. We consider the scenario where the final decision is a amalgamation of algorithm and expert opinions and deferral systems are a special case. We finally show that updating expert opinion priors with information sharing between experts is key to achieving superior performance.

## 1 INTRODUCTION

The paper discusses the challenges of facilitating communication and cooperation between machine learning models and human users in the context of healthcare. One approach to address this challenge is to prioritize interpretability and explanation in machine learning models, in particular where decision-making involves uncertainty. By creating simple models and explaining model outcomes, healthcare professionals and patients can better understand the decisions made by the model and make more informed decisions themselves (Wilson & Daugherty, 2018; Han et al., 2011). To explore this approach, we propose a Bayesian framework for human-in-the-loop systems (Wu et al., 2022) and leverage collaborative intelligence (Wilson & Daugherty, 2018). Our framework is designed to tackle these situations with ease; we demonstrate this by designing an experiment where the expert models were trained on subsets of features, mimicking the real-life experts who have domain knowledge in specific subfields of medical science. Our framework provides an optimal way to estimate the posterior probabilities for the model parameters and an optimal way to combine the inference of the experts. We also show that deferral systems (Keswani et al., 2021) are a special case where the combining expert opinions follows a categorical distribution.

## 2 WHY IS HUMAN EXPERTISE STILL RELEVANT?

In healthcare, human emotions and connections are crucial, and the patient's voice should be at the center of concern (Jeffrey, 2016). Although machine learning algorithms may outperform humans in many tasks, humans still have an advantage in multi-tasking and transfer learning. Human-in-the-loop machine learning is necessary to monitor and control algorithms, and data limitations or biases can affect the accuracy of algorithm outputs. Specialists and their domain expertise can provide valuable information beyond what is captured in the data. Qualitative data, such as in-depth interviews, can be important in understanding the etiology of a patient's condition. Rare events may not be included in a dataset, which can be problematic for rare disease diagnoses.

## 3 METHODOLOGY

In this Section, we consider a retrospective sample of 462 males in a heart-disease high-risk region of the Western Cape, South Africa (Rossouw, 1983; Hastie et al., 2009). The goal is to predict whether the patient has a coronary heart disease ("CHD"). The data was transformed by standardising the clinical metrics.

**Model**  We suppose that a patient has CHD if the patient's propensity ($s$) is high enough (i.e., $s > \tau$ for an inobservable threshold $\tau$). We refer the reader to Section A for the full mathematical formulation. Ours is a simplistic example but extensions are easy to imagine, including more experts (both human and algorithmic), more tasks, and learning. Moreover, to preserve some diversity, the mixture itself can be hierarchical. The overall learning process is best described in Figure 1.

**Scenarios**  In short, we consider one algorithm and four human experts, under five scenarios. In the first four scenarios we consider the joint decision taken by the experts and the algorithm, while in the last scenario we have a deferral system, with the system choosing to defer to one or more human experts in cases of input where the classifier has low confidence. (1) The algorithm and expert influence each other's decision during training which we refer to as *information sharing*. (2) The algorithm and expert influence each other's decision and update their beliefs as new data becomes available which we refer to as *information sharing with dynamic priors*. (3) The algorithm and expert do not influence each other during training which we refer to as *independent learners*. (4) The algorithm and expert do not influence each other but update their beliefs as new data becomes available which we refer to as *independent learners with dynamic priors*. (5) The algorithm and expert do not influence each other and a tertiary model selects an expert to infer from which we refer to as *deferral system*.

**Results**  We provide the results of our experiments in Table 1. As seen in our approach, the model performs better when information sharing is possible between the experts and the algorithm and outperforms when priors are dynamically updated.

| Methodology | AUC ROC | F1 score |
|---|---|---|
| Deferral system | 78.91 | 64.28 |
| Independent learners with dynamic priors | 75.37 | 57.63 |
| Independent learners | 72.85 | 58.82 |
| Information sharing with dynamic priors | 78.15 | **64.86** |
| Information sharing | 74.24 | **64.51** |

Table 1: Model performance

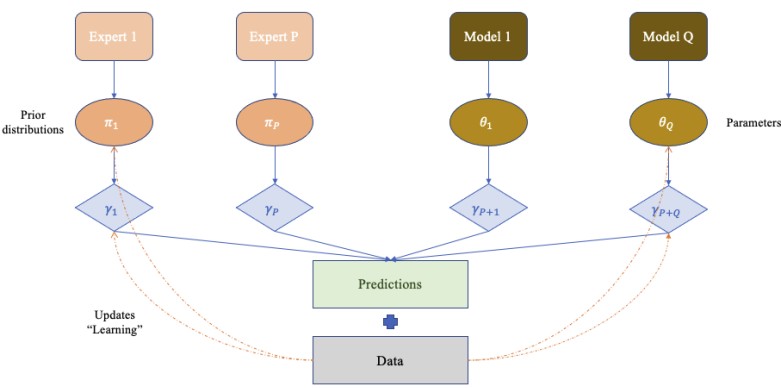

Figure 1: High-level view of human/machine collaboration with learning.

## 4  CONCLUSION

We hope to have shown, that it is possible to include human domain expertise into algorithms and estimation processes. Our experiments conclude that information sharing between the experts during estimation outperforms other scenarios and we hypothesise this corresponds to the experts learning from errors made by themselves and their peers. Managing human feedback loop for is the key to improving the quality of hybrid systems. Finally, our framework incorporates the real world scenario where experts priors update as new observations become available.

## 5 URM STATEMENT

Authors Kabir Thakur and Parth Pahwa meet the URM criteria of ICLR 2023 Tiny Papers Track.

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

## A MATHEMATICAL SET-UP

### A.1 GENERAL SETTING

Throughout this paper, $\phi_{\mu,\sigma^2}$ refers to the probability density function of a Gaussian variable with mean $\mu$ and variance $\sigma^2$ (i.e., $N(\mu, \sigma^2)$). Similarly, $\Phi$ denotes the standard Gaussian cumulative distribution function. Geometric mixtures of experts can be found in Williams et al. (2001); Hinton (2002). In this case study, we look at the problem of

- $C$ different categories, $c = 1, \cdots, C$,
- $P$ different experts, $j = 1, \cdots, P$.

**Latent model** We suppose that a user $i$ is a member of class $c$ if their propensity (or "interest") $s_i^c$ is above an (unknown) threshold, $\tau$. In other words, for every $i$ in category $c$,

$$\text{Class}_i^c = \mathbf{1}_{\{s_i^c > \tau\}}. \tag{1}$$

In addition, we assume that the latent propensity can be expressed as $s_i^c = \mu_c + \sigma_c \varepsilon_i$, where the $\varepsilon_i$'s are identically and independently distributed standard Gaussian variables. In other words, propensities are independent but their parameters are determined by the category $c$.

**Expert priors** Each expert provides a prediction on $\mu_c$ (experts do not need to have an opinion on *everything*). To be exact, we posit that each expert has a *prior* on $\mu_c$ that follows a normal distribution $N(\mu_{c,j}, \sigma_{c,j}^2)$. We also assume that the threshold, $\tau$, is common for all prospects and categories, and has a Gaussian prior: $\tau \sim N(t, v)$.

**Mixture of experts** Each expert is given a weight $\gamma_j$, which represents the importance given to his or her opinions. The vector of weights $\boldsymbol{\gamma} = (\gamma_1, \cdots, \gamma_P)'$ belongs to the simplex and has a Dirichlet prior: $\boldsymbol{\gamma} \sim \text{Dirichlet}(\boldsymbol{\alpha})$. The experts are considered independent.

**Learning** As usual, we suppose that the overall system learns (by applying Bayes' rule), so that the weights get updated as information and observations become available.

## A.2 SIMPLE CALCULATIONS

Since the mixture is geometric, the prior on each $\mu_c$, $c = 1, \cdots, C$ can be written as

$$\text{Prior}(\mu_c) \propto \prod_{j=1}^{P} \phi_{\mu_{c,j}, \sigma_{c,j}^2}(\mu_c)^{\gamma_j}, \tag{2}$$

which can be shown to be a Gaussian distribution with variance $\overline{\sigma_c^2}(\boldsymbol{\gamma})$ such that

$$\frac{1}{\overline{\sigma_c^2}(\boldsymbol{\gamma})} = \sum_{j=1}^{P} \frac{\gamma_j}{\sigma_{c,j}^2}.$$

Furthermore, its mean is $\overline{\mu_c}(\boldsymbol{\gamma})$, which is worth

$$\overline{\mu_c}(\boldsymbol{\gamma}) = \frac{\sum_{j=1}^{P} \frac{\gamma_j \mu_j}{\sigma_{c,j}^2}}{\sum_{j=1}^{P} \frac{\gamma_j}{\sigma_{c,j}^2}}.$$

## A.3 ESTIMATION

Let us now suppose that, for each category $c$, there are $n_c$ users and $a_c$ of these who are a member. Based on the data generating process, it is easy to see that the posterior distribution of the parameters $\boldsymbol{\mu} = (\mu_1, \cdots, \mu_C)'$, $\tau$ and $\boldsymbol{\gamma}$ is given by

$$\text{Posterior} \propto \prod_{c=1}^{C} \binom{n_c}{a_c} \Phi\left(\frac{\mu_c - \tau}{\sigma_c}\right)^{a_c} \Phi\left(\frac{\tau - \mu_c}{\sigma_c}\right)^{n_c - a_c} \phi_{\overline{\mu_c}(\boldsymbol{\gamma}), \overline{\sigma_c^2}(\boldsymbol{\gamma})}(\mu_c) \phi_{t,v}(\tau) \text{Dirichlet}_{\boldsymbol{\alpha}}(\boldsymbol{\gamma}). \tag{3}$$

## A.4 DYNAMIC PRIORS

We consider the scenario where at time $t + 1$ new observations have $X_{t+1}$ have become available. Consider $\theta$ is the set of model parameters. Then, using Bayes rule:

$$\text{Posterior } p(\theta | X_t, X_{t+1}) \propto p(\theta | X_t) p(X_{t+1} | \theta) \tag{4}$$

Hence learning on combined new data is equivalent to updating priors and estimating posterior only using the new obeservations.

## B EXPERIMENTATION

Multiple methods are available to compute the posterior distribution, in our example, however, we have limited ourselves to the Maximum A Posteriori ("M.A.P."), which is obtained via the L-BFGS-B optimisation method.

In addition, we have set parameters as follows:

- The threshold $\tau$'s prior is $N(0, 1)$.
- The weights' prior is Dirichlet$(1, 1, 1, 1, 1)$.
- Prior on each expert is $N(0, 1)$.

We begin by introducing the following notation. Let for user $i$, $\mathbf{x}_{i,model}$ be the observations available to the algorithm, $\boldsymbol{\beta}_{model}$, with prior $N(\mathbf{0}, \mathbf{I})$ represent the unobserved population parameters and $\mu_{i,model}$ be the log odds ratio. We know that

$$\mu_{i,model} = \mathbf{x}_{i,model}^T \boldsymbol{\beta}_{model} + \epsilon_{i,model} \tag{5}$$

where $\epsilon_{i,model}$ are are identically and independently distributed standard Gaussian variables. Similarly, for expert $j \in [1, P]$, let $\mathbf{x}_{i,expert_j}$ be the observations available to the expert, $\boldsymbol{\beta}_{expert_j}$, with prior $N(\mathbf{0}, \mathbf{I})$ represent the unobserved population parameters and $\mu_{i,expert_j}$ be the log odds ratio. We know that,

$$\mu_{i,expert_j} = \mathbf{x}_{i,expert_j}^T \boldsymbol{\beta}_{i,expert_j} + \epsilon_{i,expert_j} \tag{6}$$

where $\epsilon_{i,expert_j}$ are are identically and independently distributed standard Gaussian variables. Let $\boldsymbol{\gamma} \sim \text{Dirichlet}(\boldsymbol{\alpha})$ represent the vector of weights and $\hat{\mu}_i = [\mu_{i,model}, \mu_{i,expert_1} \cdots \mu_{i,expert_P}]$, we calculate the propensity $s_i$ as

$$s_i = \boldsymbol{\gamma}^T \hat{\mu}_i + \epsilon_i, \tag{7}$$

and $\hat{y} \sim \text{Bernoulli}(p)$, $p = \mathbf{1}_{\{s_i > \tau\}}$ where $\tau \sim N(0, 1)$.

## B.1 INFORMATION SHARING

We consider the scenario where during estimation phase, the observed information is jointly used by the experts and the algorithm to estimate the posterior distribution of the parameters. This corresponds to experiments described as (i) and (ii) in the experimentation section. The graphical representation of the model is provided in figure 2.

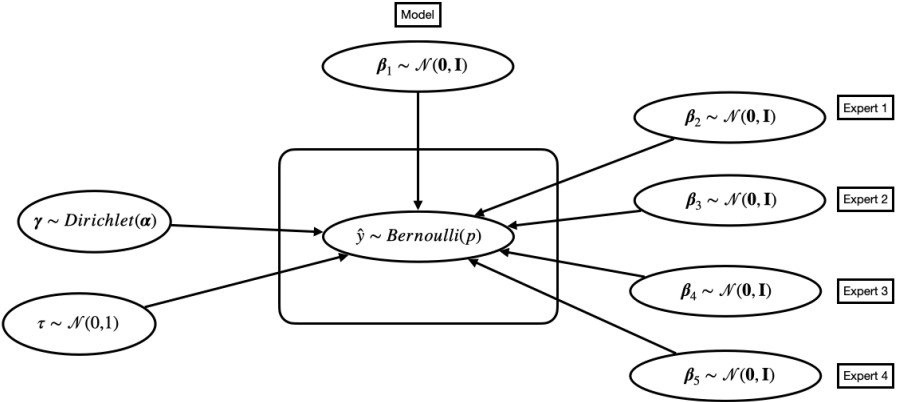

Figure 2: Information sharing model

## B.2 INDEPENDENT LEARNING

We consider the scenario where during estimation phase, the observed information is independently used by the experts and the algorithm to estimate the posterior distribution of the parameters. This is analogous to a super-learner Van der Laan et al. (2007) of the experts and the algorithm. The graphical representation of the model is provided in figure 3.

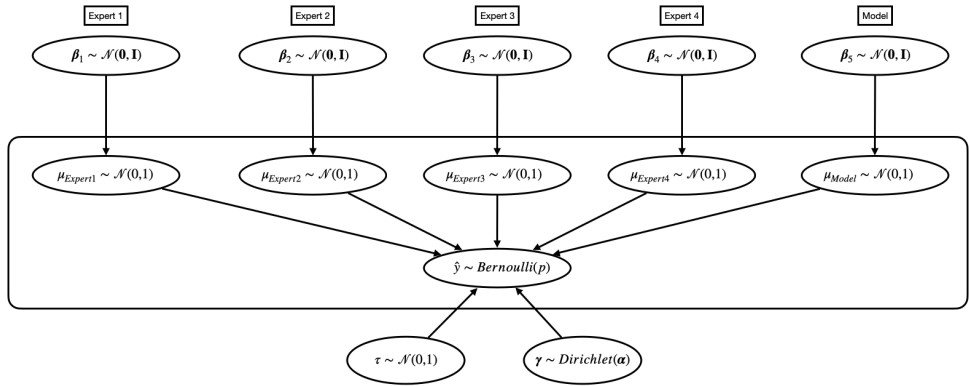

Figure 3: Independent learner model

## B.3 DEFERRAL SYSTEM

The goal of the deferral system is to defer to one or more experts who are likely to make accurate decision for the given input. We model the deferral system as a multivariate multinomial bayesian logistic regression, i.e. for customer $i$, $\gamma_i \sim \text{Categorical}(\mathbf{p}_i)$ and $\mathbf{p}_i = \text{softmax}(x_{i,model}^T \beta_{deferral,c})$ which given any user attributes, will choose the appropriate expert including the algorithm. The estimation of the posterior distributions of the parameters is done is two steps: (1) The priors of the experts and parameters of the algorithm are evaluated in an independent setting as described in B.2. The graphical representation of the model for this step is provided in figure 4.

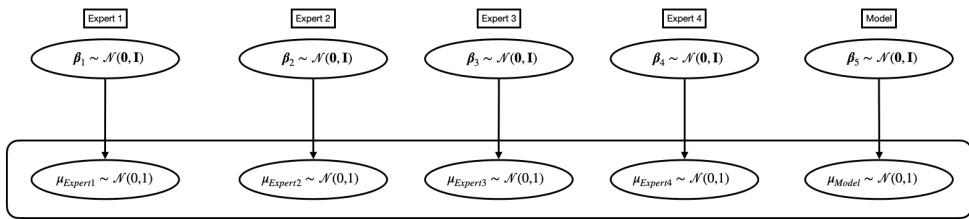

Figure 4: Deferral system: independent learners

(2) We simulate using the parameters in the previous step and assign the best predictive expert among the set for each observation. Using the augmented data set, we estimate parameters for the deferral system. The graphical representation of the model for this step is provided in figure 5

During inference, we first infer from the deferral system to pin point if we need to defer to an expert or use the algorithm, followed by the inference from the appropriate expert/algorithm.

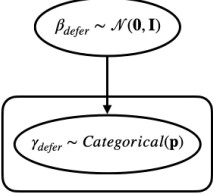

Figure 5: Deferral system: expert selection

