# OpenReview forum: "Dynamic Human AI Collaboration"
_ICLR.cc/2023/TinyPapers — Submitted to Tiny Papers @ ICLR 2023_

### Official Review · Reviewer_SLaD · 2023-03-23

**Confidence:** 3

**Summary Of Contributions:**

The authors present a scenario where "human experts" (represented by statistical models) and machine learning models share information to improve the accuracy in a classification task. This is driven by interpretability (important in the medical field) and the hypothesis that human priors can help the model to better understand the problem (and viceversa).

**Rating:**

Clear, Correct, and Reproducible (CCR): a submission which meets the reviewing criteria

**Strengths And Weaknesses:**

### Strengths
1. The paper is clear and easy to follow, even though some parts need more clarification.
1. The statistical models are formally explained.
1. The results seem to confirm the hypothesis that information sharing between experts and models improves performances.
1. The authors found an interesting way to model "human experts" through statistical models. This allows them to carry out different scenarios in their experiments.
### Weaknesses
1. Some grammatical errors (e.g "we propose a Bayesian framework **to for** [...]", "4 human expert [...]") are scattered throughout the paper.
1. The decision of $\gamma$ is not completely clear (to me), even though, intuitively, it would seem an important parameter. Each expert is only trained on a subset of the information available, does $\gamma$ depend on the "amount" of data known by each expert (how much of an "expert" they are)? From the appendix I concluded it is randomly rapresented by a probability distribution, but further explanation would be nice.

**Suggested Changes:**

1. Add the results of a simulation with "experts only". This is driven by the fact that, in the "Deferral system", we see a huge jump in performance compared to the "independent learners". The ablation study would confirm (or deny) that the model is actually improving given the experts opinions, and is not just using them to make the decision (by setting the experts' weight $\gamma$ higher than the model's).
1. Better explain and justify the $\gamma$ weights. How would they translate in the "real-world" scenario?

---

### Official Review · Reviewer_coED · 2023-04-02

**Confidence:** 3

**Summary Of Contributions:**

This paper proposes to improve the performance of Machine Learning models by sharing information with experts in the problem domain. The authors have conducted an experiment using a real-world dataset to demonstrate the proposed method, and the results indicate that sharing information with experts indeed causes improved performance.

**Rating:**

High Potential (HP): a submission which meets the reviewing criteria and has potential to make an impact on the field

**Strengths And Weaknesses:**

Strengths of the paper

* The proposed idea to incorporate expert opinions to improve the performance of the machine learning models is exciting and timely.

* Overall, the paper is clearly written, and the claims are backed up with empirical results.

* The information given in the main text is not sufficient to reproduce the results claimed in the paper, but I believe the supplementary materials serve this purpose.

* The paper follows the basic requirements of the conference.

 Weaknesses of the paper

* The paper has not discussed the related work and where this paper stands w.r.t the existing work. The authors are encouraged to highlight the research gap they try to fulfill in this work.

* The paper can be improved by expanding the Methodology section with more details of the proposed method, at least with high-level details.

* It would be better if the author could discuss why AUC ROC and F1 Score disagree.


**Suggested Changes:**

Please check the strengths and weaknesses section.

---

### Author Response · Authors · 2023-06-01
**Opt-in for archival**

The authors opt in for archival of the tiny paper.

---

### Meta-Review · Area_Chair_UbCC · 2023-04-04

**Recommendation:** Invite to present
**Confidence:** 4

**Metareview:**

The reviewers agree that this paper is mostly well-written, and the method is interesting with improvements shown in the experiments. The paper may be improved by more clearly discussing the method and the difference with related works.


**Summary:**

This work presents a scenario where machine learning models and human experts share information and collaborate to improve the performance.

**Reason For Not Giving A Higher Recommendation:**

Lack of discussion on related works and lack of some methodology details.


**Reason For Not Giving A Lower Recommendation:**

The method is interesting with improvements shown by the experiments.

---

### Decision · Program_Chairs · 2023-04-07

Invite to present